# Fund style drift and fund performance: Evidence from China

Yaozhi Chen[1], Honghong Wei[2]*

1 School of Finance, Shanghai University of Finance and Economics, Shanghai, P.R. China, 2 College of Business, Shanghai University of Finance and Economics, Shanghai, P.R. China

* weihonghong@163.sufe.edu.cn

**Data Availability Statement:** All data are available from kaggle at https://doi.org/10.34740/KAGGLE/DSV/10065916.

**Funding:** This research was funded by the National Natural Science Foundation of China (No. 72073086). The funders had no role in study

## Abstract

This study selects the quarterly data of all equity and equity-oriented hybrid open-end funds in China from 2007 to 2022 as the research sample, and examines the impact of fund style drift on fund returns through a two-ways fixed effect model. Our results show that overall style drift tends to improve fund performance. However, after differentiating the type of style drift, we find that fund drift based on stock picking abilities enhanced performance, whereas fund drift based on chasing market trends reduced fund performance. This study presents a new measurement method based on industry allocation, providing an empirical foundation for research on industry-specific theme funds. It also offers fund managers insights for optimizing performance evaluation and incentive systems, while serving as a reference for regulators to develop flexible, effective policies for market stability.

## 1. Introduction

As of the second quarter of 2024, China's active equity funds totaled 3,650, with an aggregate size of 3.26 trillion yuan. The continuous growth of these funds reflects the rising demand for wealth management among Chinese citizens. Fund investment style is a crucial determinant of performance, helping investors assess risks and returns. However, due to changes in market conditions and the performance pressures faced by fund managers, many funds gradually deviate from their declared investment styles, a phenomenon known as style drift. Numerous studies have shown that a fund's investment style can evolve over time, deviating from the investment direction, industry scope, and stock selection criteria outlined in its fund contract [1–4]. Kim et al. [2] found that only 46% of funds maintained their investment style unchanged. In 2021, for example, at least 29 non-consumption and non-liquor-themed funds in China held Guizhou Maotai in their top ten holdings. In response, the China Securities Regulatory Commission (CSRC) issued documents such as the 2022 Opinions on Accelerating the High-Quality Development of the Public Fund Industry and the Institutional Supervision Bulletin, which emphasized limiting fund style drift, high turnover, and short-term trading behaviors aimed at earning quick profits. However, a series of studies have indicated that style drift can actually enhance fund performance [5–7]. If style drift can improve performance, why would regulators want to restrict it? Should style drift be strictly controlled? The impact of style drift on fund performance remains an important issue that warrants further exploration.

design, data collection and analysis, decision to publish, or preparation of the manuscript.

**Competing interests:** The authors have declared that no competing interests exist.

Investment style helps categorize assets, concentrating investments in specific categories. This not only allows investors to select funds based on their risk preferences and return expectations, but also significantly reduces the cost of gathering information [8]. A stable and predictable investment style provides investors with a clear reference point, while also offering fund managers a pathway to achieve consistent returns and manage risks [9]. However, research has shown that a fund's investment style is not always consistent. Over time, funds may deviate from their initially established investment styles [10–11]. This drift can partially reflect a fund manager's active management ability, but it may also lead to discrepancies between investor expectations and actual performance [12]. A typical example of this in China is the Qianhai Kaiyuan Utilities Fund, which was one of the top performers in 2021. Despite being classified as a utilities fund, it was heavily invested in new energy companies, sparking market concerns about the authenticity of its style and providing a valuable opportunity to study how style drift affects fund performance.

To examine the specific impact of style drift on fund performance, this study approaches the issue from the perspective of industry allocation, analyzing the decision-making differences driven by various motivations behind style drift. This method allows for a deeper understanding of fund managers' investment strategies in dynamic market environments and how these strategies influence overall fund performance. Unlike previous studies, this research measures style drift by comparing the industry allocation differences between the fund and its performance benchmark [13]. The results indicate that, overall, style drift tends to improve fund performance. However, trend-chasing drift has a negative impact on performance, while investment ability-driven drift positively affects performance. The latter occurs because skilled fund managers, who possess strong market judgment and analytical abilities [14], have an information advantage in navigating market shifts [6, 15] and can generate excess returns through stock selection and timing strategies [16]. However, style drift is not always based on rational analysis. It can also result from fund managers' herd behavior [15], where trend-chasing drift increases trading costs [4] and the potential for misjudgments and decision errors [17], ultimately leading to a decline in performance [18].

This study may make contributions in the following aspects. Firstly, while prior literature focuses on the overall impact of fund style drift, this study differentiates the effects of distinct types of fund style drift behaviors on fund performance. This study explores the impact of style drift based on fund investment ability and trend-chasing style drift on performance, offering new insights into the economic consequences of style drift controversies. Secondly, past research often categorized fund styles based on market capitalization (large, medium, small) and growth/value characteristics [10]. In contrast, this study emphasizes industry classification, as investors frequently obtain information about a fund's industry classification. The study offers an approach for quantifying style drift in industry-themed funds. Lastly, the study of the impact of style drift on fund performance not only enriches the literature on the economic consequences of style drift, but also examines whether style drift protects the interests of investors, deepens our understanding of the value of active management of funds, guides investors to rationally analyze popular funds in the market, and treat fund style drift with caution.

## 2. Literature review and hypotheses

Style drift can be seen as a strategy of active fund management, where portfolio managers adjust the weight of different investment styles in search of higher returns [19]. A large body of literature has found that the investment style of funds often does not match their stated investment objectives [1–3], as funds deviate from the investment directions, style characteristics,

industry scope, and stock selection criteria defined in their fund contracts [11]. This behavior, where the fund's style characteristics change over time, is referred to as style drift [19].

There is ongoing debate in the literature regarding the relationship between style drift and fund performance. Some studies suggest that style drift can enhance fund performance. For example, Sha [7] used a semi-annual style drift indicator to examine the relationship between style drift and the performance of Chinese mutual funds, finding that style drift helped improve the performance of Chinese funds. Andreu et al. [6] found that style drift leads to better performance, especially for small-cap, growth, and high-alpha funds, even after accounting for additional transaction costs [5]. In contrast, other studies argue that style drift weakens fund performance. Brown and Zhang [1] found that maintaining style consistency leads to better absolute and relative performance. Mateus et al. [3] found no evidence that funds with style changes performed better than those without such changes. Yi et al. [18] found a negative correlation between style drift and fund performance.

As a fund management strategy, style drift can enhance returns by adjusting the weight of different style stocks in the portfolio [17, 20]. Style drift can effectively leverage a fund manager's stock selection and timing abilities. Fund managers can flexibly adjust between different investment styles, maximizing opportunities and generating higher returns in changing market conditions. Moreover, Avramov and Wermers [20] showed that funds can earn 2% to 4% excess returns annually through industry rotation timing during business cycles. Allocating to booming industries can yield 3% to 6% excess returns, suggesting that style drift strategies can generate significant excess returns for funds.

Style drift also attracts more investors and drives asset prices up. When a particular style performs well, it draws more investors, thus pushing up the prices of those assets [8]. This positive feedback loop from investor sentiment not only drives asset prices higher but also creates more investment opportunities for the fund. Chua and Tam [17] found that funds with style drift could achieve excess returns in the short term, constantly attracting capital inflows and forming a capital accumulation effect. Based on these analyses, we propose the following hypothesis:

**Hypothesis 1**: *The stronger the degree of fund style drift, the better the fund performance.*

Style drift, as a dynamic investment strategy, places higher demands on the ability of fund managers. Successfully implementing style drift requires not only the ability to recognize market trends and analyze market conditions but also the ability to make sound investment decisions and manage risks. Fund management teams need to maintain sharp judgment in changing market environments and adjust their portfolios in a timely manner to respond to different market cycles and economic conditions, thereby achieving higher investment returns.

Existing research shows that a fund's ability has a significant impact on its performance, and the effectiveness of the style drift strategy is closely related to the fund manager's analysis, judgment, timing, and risk management abilities [9]. Many opportunities for style rotation exist in the market, but those who profit are often the funds with informational advantages [6]. Kacperczyk et al. [21] pointed out that skilled fund managers tend to concentrate their holdings in industries where they have informational advantages, thus generating excess returns during market fluctuations. Sun et al. [14] found that the stronger a fund's independent decision-making ability, the better its future performance. This informational advantage enables the fund to mobilize capital at the right time and optimize the risk-return structure of the portfolio. Huang [16] also found that less capable funds are more likely to adjust their portfolios to shift risks, whereas funds with strong stock-picking or timing abilities adjust their portfolios to achieve superior performance. Therefore, the fund manager's ability to process information

and select stocks plays an important role in style drift. Based on this, we propose the following hypothesis:

**Hypothesis 2**: *Funds with higher investment ability will achieve higher performance when engaging in style drift.*

Although style drift strategies can help funds adapt to changing market conditions and seek excess returns, they are not without risks. If fund managers fail to accurately predict market trends or blindly follow the investment styles of high-performing funds, it often leads to trend-chasing behavior, which increases the investment risk of the fund. Trend-chasing behavior refers to fund managers chasing sectors or stocks that have already experienced significant price increases in the market. This behavior can lead to investment decisions that deviate from the fund's long-term objectives and increase the non-systematic risk of the fund. According to behavioral finance theory, when funds lack sufficient investment analysis capability, they are more prone to irrational behavior. Common irrational investment behaviors under performance pressure include herd behavior and buying high and selling low. Buying high and selling low refers to fund managers exhibiting a higher risk appetite for stocks that have risen in price and risk aversion for stocks that are underperforming. This preference can lead to a shift in the risk profile of the fund's investment portfolio.

Implementing a style drift strategy requires a comprehensive consideration of market trends, fundamental analysis, and risk management. In contrast, trend-chasing behavior focuses more on short-term market profits, leading to frequent trading. Brown et al. [1] pointed out that excessive focus on short-term profits through trend-chasing behavior increases the likelihood of decision errors and judgment mistakes, thus raising the non-systematic risk of the fund. At the same time, investors often use short-term returns to evaluate fund managers' abilities. If a fund performs poorly in the short term, investors may quickly withdraw their capital. This liquidity pressure forces fund managers to adopt a short-term perspective, which in turn affects their long-term investment decisions [16]. Furthermore, opportunistic trading behavior driven by trend-chasing increases the fund's transaction costs, limits its investment opportunities, and weakens overall performance [4, 18]. Based on these analyses, we propose the following hypothesis:

**Hypothesis 3**: *Funds style drift based on market trends will reduce performance when engaging in style drift.*

Through a comprehensive review of domestic and international literature, this study finds that research on style drift at the industry allocation level is still in its early stages, with a lack of systematic analytical frameworks and measurement methods, particularly regarding the mechanisms of dynamic industry adjustments on style drift. Most existing literature focuses on drift measurements based on market capitalization or style categories [10], with limited exploration of the impact of dynamic industry changes on fund performance. Additionally, existing research shows significant disagreement on the impact of style drift on fund performance, with insufficient consideration of the heterogeneity of fund manager motivations [15–18].

To address these gaps, this study innovatively proposes a style drift measurement method based on the industry allocation ratio of funds. This method quantifies style drift by comparing the deviation between the fund's holdings and the industry allocation of its performance benchmark [13], offering a more precise reflection of changes in fund style. Furthermore, this study distinguishes between two types of style drift: investment ability-driven drift and trend-chasing drift. By analyzing their heterogeneous effects, we not only explain the discrepancies in the literature but also uncover the complexity of how style drift affects fund performance, providing a new theoretical perspective and empirical basis for the field.

## 3. Methodology

### 3.1. Data

The data utilized in this study, encompassing fund characteristics such as fund returns, fund holdings ratios, and fund sizes, as well as stock characteristics, were sourced from the WIND database. The proportion of fund benchmark holdings was extracted from the RESSET database. The Chinese fund market is highly influenced by policy, with regulatory bodies such as the China Securities Regulatory Commission (CSRC) issuing a series of policy documents to regulate style drift and market behavior. These policy interventions provide a unique backdrop for studying fund style drift and its impact. Therefore, this study focuses on quarterly data from actively managed open-end equity and equity-oriented mixed funds in China from 2007 to 2022. The choice of 2007 as the starting point is twofold: first, the number of open-end equity funds in China was limited before 2007; second, fund disclosure practices were less transparent prior to this year. Samples were eliminated with missing data related to benchmark performance and holdings ratios, resulting in a final dataset comprising 1839 funds. To mitigate the influence of outliers, the study winsorizes continuous variables at the 1% and 99% levels.

### 3.2. Variables

**(1) Fund performance.** The dependent variable in this study is fund performance. Given the absence of a significant monthly "momentum effect" in the Chinese A-share market, the Carhart four-factor model [22] was not utilized. Instead, this study employed various risk-adjustment models, including original excess return, CAPM, the Fama-French three-factor model, and the Fama-French five-factor model [23–25], to assess fund performance. Specifically, the study estimated the excess returns for each fund using monthly data spanning 2007 to 2022. Taking the Fama-French three-factor model as an example, the regression model is expressed as follows:

$$R_{it} - R_{ft} = \alpha_i + \beta_1(R_{mt} - R_{ft}) + \beta_2 SMB_t + \beta_3 HML_t + \varepsilon_{it} \tag{1}$$

Where $R_{it}$ is the return of fund $i$ in month $t$, $R_{ft}$ is the risk-free rate. This study chooses the three-month fixed deposit benchmark interest rate as the risk-free interest rate. $R_{mt}$ is the monthly market return. SMB and HML are the size factor and the book-to-market ratio factor, respectively. All relevant data come from the CSMAR database. The intercept term $\alpha_i$ is the monthly average return. This study can obtain monthly time-series data for the excess returns of each fund.

Then this study adjusts the monthly average return to quarterly, consistent with the frequency of fund style drift data. We use the cumulative excess return calculation method to calculate quarterly excess returns:

$$\alpha_{in} = (1 + \alpha_{i,3n-2})(1 + \alpha_{i,3n-1})(1 + \alpha_{i,3n}) - 1 \tag{2}$$

Where $\alpha_{in}$ is the risk factor-adjusted excess return of fund $i$ in the $n$th quarter, and $\alpha_{i,3n}$ is the risk factor-adjusted excess return of fund $i$ in the $3n$th month.

**(2) Fund style drift.** Fund style drift is the explanatory variable of this article. Buncic et al. [13] interpret the Euclidean distance between a fund's holdings portfolio and the benchmark as an indicator of portfolio activity for a specific style. This "activity" represents the degree of deviation of the fund from the style benchmark. This article refers to Buncic et al. [13] and uses the Euclidean distance between the fund and the benchmark to measure the degree of

fund style drift. The specific measurement methods are as follows:

$$SD_{F,t} = \sqrt{\sum_{j=1}^{N} \left( w_{F,j,t} - w_{M,j,t} \right)^2} \tag{3}$$

Where $w_{F,j,t}$ is the weight of industry j in the fund at time t, $w_{M,j,t}$ is the weight of industry j in the fund performance benchmark at time t, and N is the number of industries held by the fund. Due to funds disclosing their top ten holdings only quarterly, this study examines the degree of style drift within the top ten weighted stocks of the funds.

Variants of this index have also been used by Kacperczyk et al. [21]. The difference is that their variant replaces $w_{M,j,t}$ with the market portfolio instead of the benchmark portfolio. They then compare each portfolio to a market portfolio proxy such as the Russell 3000. Kacperczyk et al. [21] interpreted its Euclidean distance as industry concentration. Industry concentration measures the fund's overall industry concentration risk and whether the fund tends to invest concentratedly in certain industries. This article focuses more on whether the fund can obtain excess returns through industry allocation that deviates from its performance benchmark.

The industry classification uses the China Securities Regulatory Commission (CSRC) industry classification, which categorizes stocks into 19 industries. Manufacturing companies constitute over 60%, encompassing various sectors such as agriculture, real estate, transportation, and education. A broad categorization of stocks into manufacturing would diminish the overall drift, as changes within this rough category would be overlooked and not considered part of the drifting behavior. Therefore, this study uses the secondary classification of the CSRC industry to further categorize manufacturing stocks, to obtain a more precise measure of drift.

**(3) Control variables.**   To control for other factors affecting fund performance, this study incorporates control variables related to fund characteristics and fund manager characteristics into the regression model. The specific calculation methods for these variables are as follows:

This study selects several fund characteristics as control variables, as fund size and fund age are known to influence fund performance [26]. Therefore, fund size (Size) is measured as the natural logarithm of the total assets of the fund, and fund age (Fundage) is calculated as the current year minus the year the fund was established. Additionally, since concentrated investors can take advantage of information asymmetry to achieve higher performance [27], this study also controls for investment concentration (Conc), which is calculated as the weighted average of the ratio of shares held by the fund's top ten positions to the circulating shares of each stock. Since changes in fund managers can also impact the fund's portfolio, we control for the fund manager change (Change) variable. This is a dummy variable that takes the value of 1 if the fund manager changes during the quarter, and 0 otherwise.

Considering that fund manager characteristics may also influence fund performance [28], this study controls for various fund manager-level variables. These include fund manager gender (Gender), which takes the value of 1 if the manager is male and 0 otherwise; fund manager education (Edu), where a value of 1 corresponds to less than a bachelor's degree, 2 corresponds to a bachelor's degree, 3 corresponds to a master's degree, and 4 corresponds to a PhD or higher; fund manager work experience (Work), which represents the number of years the manager has worked in the investment industry; fund manager tenure (Serv), which represents the number of years the manager has been managing the fund; and fund manager age (Age), which corresponds to the manager's age in the current year.

In addition, for the panel data regression analysis, this study also controls for fund dummy variables (Fund) and quarter dummy variables (Quarter) to account for individual fixed effects and time fixed effects. To mitigate the influence of outliers on the regression results, all

continuous variables were winsorized at the 1% and 99% levels prior to regression. The abbreviations, definitions, and brief measurement methods for the main variables in this section are provided in Table 1.

### 3.3. Research model

To study the impact of fund style drift on fund performance, after controlling relevant variables, this study uses fund style drift to regress the fund's original excess return, CAPM alpha, three-factor alpha, and five-factor alpha respectively, and constructs an empirical model as follows:

$$\alpha_{it} = \beta_0 + \beta_1 SD_{it-1} + \beta_2 Controls_{it-1} + Quarter_t + Fund_i + \varepsilon_{it} \tag{4}$$

Where the independent variable $\alpha_{it}$ is measured by the fund's original excess return, CAPM alpha, three-factor alpha, and five-factor alpha respectively. The dependent variable $SD_{it-1}$ is the degree of fund style drift. $Controls_{it-1}$ including fund characteristic variables, fund manager replacement variables, and fund manager characteristic variables, which are specifically defined as described above. In addition to controlling variables, this study also controls fund fixed effects $Fund_i$ and time fixed effects $Quarter_t$.

## 4. Style drift and fund performance

### 4.1. Descriptive statistics

Table 2 shows the descriptive statistical results of the research sample in this study. Panel A of Table 2 shows that the standard deviation of the fund's original excess return is 0.134, which is

**Table 1. Variable definitions.**

| Variable | Definition | Method |
|---|---|---|
| **Dependent Variables** | | |
| *Rp-Rf* | Raw Excess Return | The excess return of the fund relative to the risk-free rate |
| *CAPM alpha* | CAPM Excess Return | The excess return calculated by the difference between the asset's return relative to the risk-free rate and the market return |
| *FF3 alpha* | Fama-French Three-Factor Excess Return | The excess return adjusted by the market return, size factor, and value factor |
| *FF5 alpha* | Fama-French Five-Factor Excess Return | The excess return calculated by adding profitability and investment factors to the three-factor model |
| **Independent Variable** | | |
| *SD_Euc* | Fund Style Drift | The Euclidean distance between the fund's industry allocation and its performance benchmark's industry allocation |
| **Control Variables** | | |
| *Size* | Fund Size | The natural logarithm of the fund's total assets |
| *Fundage* | Fund Age | The difference between the current year and the year the fund was established |
| *Conc* | Investment Concentration | The weighted average of the ratio of shares held by the fund's top ten positions to the circulating shares of each stock |
| *Change* | Fund Manager Change | A dummy variable that equals 1 if the fund manager changed during the quarter, and 0 otherwise |
| *Gender* | Fund Manager Gender | Equals 1 if the fund manager is male, and 0 otherwise |
| *Edu* | Fund Manager Education | Equals 1 for below bachelor's degree, 2 for bachelor's degree, 3 for master's degree, and 4 for PhD or above |
| *Work* | Fund Manager Work Experience | The number of years the fund manager has worked in the investment industry. |
| *Serv* | Fund Manager Tenure | The number of years the fund manager has been managing the fund |
| *Age* | Fund Manager Age | The age of the fund manager in the current year |

**Table 2. Descriptive statistics.**

**Panel A: Full sample**

| Variables | Obs | Mean | Std. | Min | Median | Max |
|---|---|---|---|---|---|---|
| Rp-Rf | 20547 | 0.032 | 0.134 | -0.256 | 0.028 | 0.388 |
| CAPM alpha | 20547 | 0.070 | 0.431 | -0.710 | 0.015 | 1.721 |
| FF3 alpha | 20547 | 0.029 | 0.379 | -0.753 | -0.008 | 1.532 |
| FF5 alpha | 20547 | -0.002 | 0.359 | -0.784 | -0.027 | 1.405 |
| SD | 20547 | 0.042 | 0.036 | 0.001 | 0.032 | 0.197 |
| Size | 20547 | 20.054 | 1.661 | 15.592 | 20.209 | 23.205 |
| Fundage | 20547 | 6.020 | 4.083 | 1.107 | 4.737 | 21.167 |
| Conc | 20547 | 0.320 | 0.590 | 0.000 | 0.090 | 6.723 |
| Change | 20547 | 0.185 | 0.388 | 0.000 | 0.000 | 1.000 |
| Gender | 20547 | 1.069 | 0.254 | 1.000 | 1.000 | 2.000 |
| Edu | 20547 | 2.956 | 0.260 | 2.000 | 3.000 | 4.000 |
| Work | 20547 | 14.803 | 2.962 | 4.400 | 14.700 | 30.000 |
| Serv | 20547 | 6.965 | 2.126 | 0.300 | 6.800 | 18.400 |
| Age | 20547 | 45.012 | 2.953 | 31.000 | 46.000 | 60.000 |

Panel B: Difference test between groups

| Variable | High style drift | | | | Low style drift | | | Diff. |
|---|---|---|---|---|---|---|---|---|
| | Obs | Mean | Std. | | Obs | Mean | Std. | |
| Rp-Rf | 10289 | 0.035 | 0.140 | | 10258 | 0.029 | 0.128 | 0.006*** |
| CAPM alpha | 10289 | 0.082 | 0.459 | | 10258 | 0.058 | 0.401 | 0.024*** |
| FF3 alpha | 10289 | 0.035 | 0.403 | | 10258 | 0.023 | 0.353 | 0.011** |
| FF5 alpha | 10289 | 0.003 | 0.383 | | 10258 | -0.007 | 0.334 | 0.010** |
| Size | 10289 | 20.140 | 1.637 | | 10258 | 19.967 | 1.681 | 0.173*** |
| Fundage | 10289 | 6.005 | 4.092 | | 10258 | 6.035 | 4.074 | -0.031 |
| Conc | 10289 | 0.389 | 0.648 | | 10258 | 0.251 | 0.516 | 0.138*** |
| Change | 10289 | 0.176 | 0.381 | | 10258 | 0.194 | 0.396 | -0.018*** |
| Gender | 10289 | 1.049 | 0.216 | | 10258 | 1.090 | 0.286 | -0.041*** |
| Edu | 10289 | 2.961 | 0.257 | | 10258 | 2.952 | 0.263 | 0.009** |
| Work | 10289 | 14.818 | 2.898 | | 10258 | 14.789 | 3.024 | 0.029 |
| Serv | 10289 | 7.010 | 2.130 | | 10258 | 6.920 | 2.121 | 0.090*** |
| Age | 10289 | 45.014 | 2.982 | | 10258 | 45.010 | 2.924 | 0.004 |

***, **, * indicate significance at the 1%, 5%, and 10% significance levels respectively.

smaller than the fund return rate after adjustment by the CAPM, Fama-French three-factor and Fama-French five-factor models, indicating that the risk-adjusted fund return Rate volatility is greater. The average value of style drift is 0.042 and the standard deviation is 0.036, which shows that the level of style drift of fund presidents in the market is relatively large, but the difference in style drift is not significant.

Furthermore, this study divides the sample into a low style drift group and a high style drift group based on the median fund style drift. The specific results are shown in Panel B of Table 2. This study finds that the average original excess return, CAPM alpha, three-factor alpha, and five-factor alpha of funds with high style drift are higher than those of the low style drift group. The difference between average original excess return and CAPM alpha is significant at the 1% level, and the difference between three-factor alpha and five-factor alpha is significant at the 5% level. Moreover, the standard deviations of fund return in the high style drift

group are larger than those in the low style drift group, indicating that the returns in the high style drift group are more volatile.

The Panel B in Table 2 shows that the excess returns of the high style drift group are generally higher than the excess returns of the low style drift group, but this difference may also come from other factors such as risk. Therefore, based on the median annual style-drifting values, this study constructs two investment portfolios: the high style-drifting fund group and the low style-drifting fund group, to further examine the reasons for return differences.

Fig 1 reports the excess cumulative returns from the second quarter of 2008 to the second quarter of 2022 after constructing the two portfolios. Observing the time series characteristics of cumulative returns after portfolio construction, this study found that: first, the high style drift group achieved significant cumulative excess returns after the fourth quarter of 2014, while the low style drift group achieved first in 2015. Significant cumulative excess returns were only achieved after the quarter, which shows that style drift does not achieve cumulative excess returns at all times; secondly, the portfolio with high style drift achieved higher returns than the portfolio with low style drift. excess returns, suggesting that style drift has a greater impact on fund performance. This is because style drift effectively leverages the fund manager's stock selection and timing abilities. Fund managers can flexibly adjust between different investment styles, maximizing opportunities and generating higher returns in response to changes in market conditions [20].

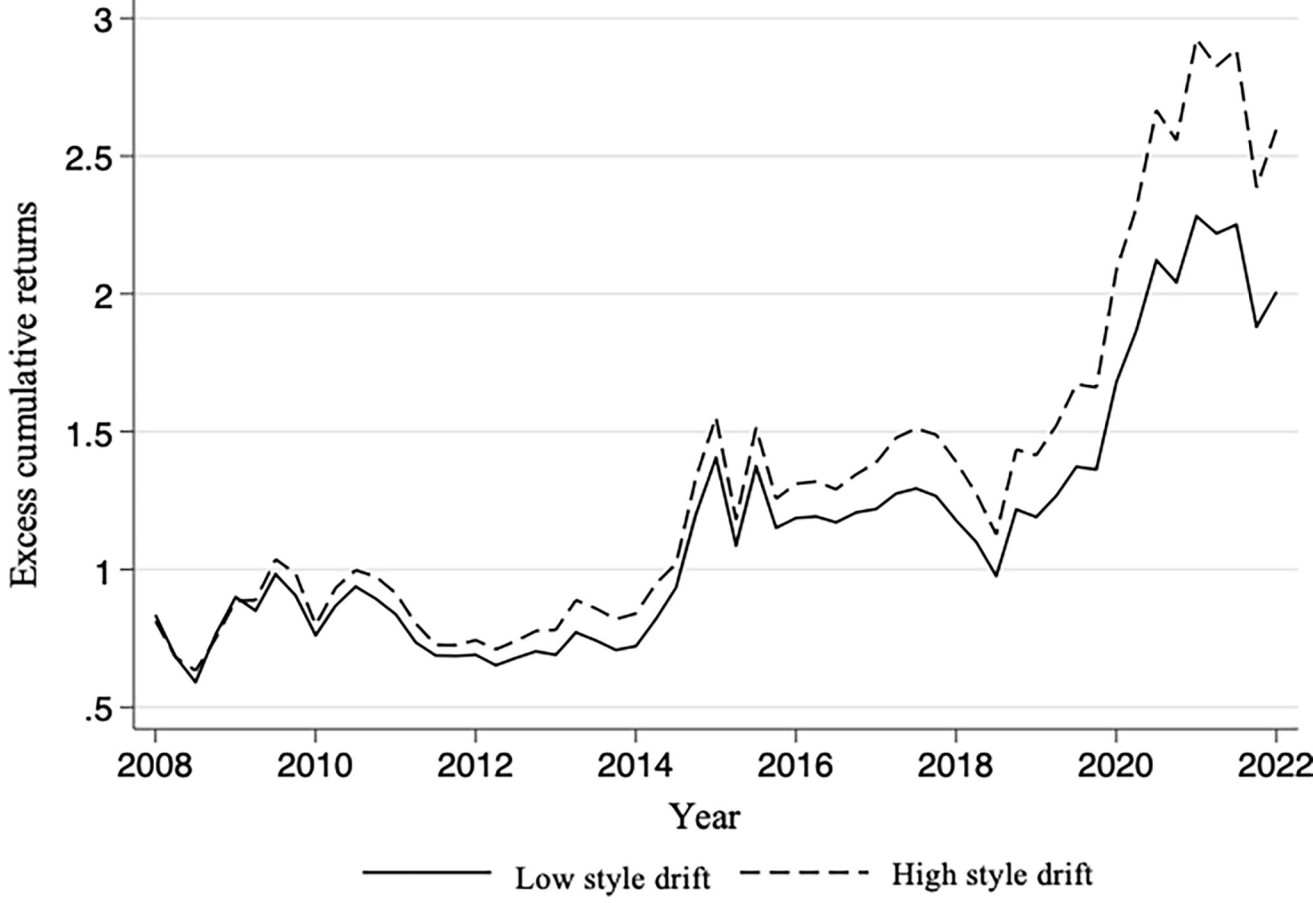

**Fig 1. Excess cumulative returns on investment portfolio.**

## 4.2. Regression results

To test the overall impact of fund style drift on fund performance, the fund performance in regressions (1) to (4) is measured by original excess return, CAPM alpha, three-factor alpha, and five-factor alpha respectively. This study uses fund Style drift to regress fund performance. Table 3 shows the baseline results of the regression. Column (1) of Table 3 uses original excess returns to measure fund performance as the independent variable. In the regression, the coefficient of the dependent variable that this study is most concerned about is the regression coefficient of the style drift SD index, which is 0.113. This means that for every standard deviation increase in the style drift index (equivalent to 3.6%), the fund's quarterly original return will increase by 0.4 basis points (0.113*0.036), or approximately 0.016% on an annual basis. This increase accounts for the original return. The rate is 12.7% of the average. This coefficient is statistically and economically significant.

Table 3 uses CAPM alpha to measure fund performance as the independent variable. The regression coefficient of the SD index is 0.366. This means that for every one standard deviation increase in the style drift index, the fund's CAPM alpha will increase by 1.32 basis points (0.366*0.036), or approximately 0.053% on an annual basis. This increase accounts for 18.8% of the average CAPM alpha. This coefficient is statistically and economically significant. Columns (3) and (4) of Table 3 use three-factor alpha and five-factor alpha respectively to measure fund performance as the independent variables. The regression results are consistent.

**Table 3. Style drift and fund performance.**

|  | (1) | (2) | (3) | (4) |
|---|---|---|---|---|
|  | **Rp-Rf** | **CAPM alpha** | **FF3 alpha** | **FF5 alpha** |
| SD | 0.113*** | 0.366*** | 0.273*** | 0.303*** |
|  | (5.81) | (3.66) | (2.95) | (3.30) |
| Size | 0.002** | 0.008** | 0.015*** | 0.009** |
|  | (2.33) | (2.12) | (4.30) | (2.41) |
| Fundage | -0.020*** | -0.036** | -0.027* | -0.019 |
|  | (-6.21) | (-2.19) | (-1.76) | (-1.23) |
| Conc | -0.000 | 0.002 | -0.033*** | -0.026*** |
|  | (-0.27) | (0.30) | (-5.32) | (-4.21) |
| Change | -0.005*** | -0.033*** | -0.024*** | -0.023*** |
|  | (-3.75) | (-4.56) | (-3.66) | (-3.44) |
| Gender | -0.012** | -0.053** | -0.042* | -0.045** |
|  | (-2.51) | (-2.15) | (-1.83) | (-1.98) |
| Edu | -0.000 | -0.007 | -0.011 | -0.005 |
|  | (-0.01) | (-0.29) | (-0.49) | (-0.23) |
| Work | -0.001* | -0.004 | 0.000 | 0.003 |
|  | (-1.81) | (-1.35) | (0.14) | (0.99) |
| Serv | -0.000 | 0.002 | -0.000 | -0.002 |
|  | (-0.03) | (0.55) | (-0.04) | (-0.63) |
| Age | 0.001 | 0.005 | -0.003 | 0.003 |
|  | (0.55) | (0.78) | (-0.52) | (0.45) |
| Fund FE | Yes | Yes | Yes | Yes |
| Time FE | Yes | Yes | Yes | Yes |
| $N$ | 20547 | 20547 | 20547 | 20547 |
| $R^2$ | 0.773 | 0.395 | 0.289 | 0.223 |

***, **, * indicate significance at the 1%, 5%, and 10% significance levels respectively, and the t statistics are in parentheses.

**Table 4. Robustness test.**

| Panel A: Manhattan distance | | | | |
|---|---|---|---|---|
| | (1) | (2) | (3) | (4) |
| | Rp-Rf | CAPM alpha | FF3 alpha | FF5 alpha |
| SD_new | 0.031*** | 0.120*** | 0.133*** | 0.104*** |
| | (6.03) | (4.59) | (5.47) | (4.31) |
| Controls | Yes | Yes | Yes | Yes |
| Fund FE | Yes | Yes | Yes | Yes |
| Time FE | Yes | Yes | Yes | Yes |
| $N$ | 20547 | 20547 | 20547 | 20547 |
| $R^2$ | 0.774 | 0.395 | 0.289 | 0.223 |
| Panel B: Benchmark-adjusted excess returns | | | | |
| | Rp-Rb | CAPM alpha[b] | FF3 alpha[b] | FF5 alpha[b] |
| SD | 0.099*** | 0.576*** | 0.276*** | 0.307*** |
| | (5.38) | (4.53) | (2.96) | (3.32) |
| Controls | Yes | Yes | Yes | Yes |
| Fund FE | Yes | Yes | Yes | Yes |
| Time FE | Yes | Yes | Yes | Yes |
| N | 20547 | 20543 | 20543 | 20543 |
| $R^2$ | 0.444 | 0.318 | 0.287 | 0.221 |

***, **, * indicate significance at the 1%, 5%, and 10% significance levels respectively, and the t statistics are in parentheses.

Table 3 presents regression analyses of the relationship between fund style drift and fund performance. Across measures of fund performance, including original excess returns, CAPM alpha, three-factor alpha, and five-factor alpha, the regression coefficients for the style-drifting standard deviation (SD) index are all significantly positive. These coefficients hold both statistically and economically significant implications. These findings suggest that, overall, fund style drift enhances fund performance. This may be because style drift not only helps fund managers to exercise their ability to select stocks and timing in market changes [20], but also attracts more investors to enter the market, thus forming a positive feedback loop and further promoting the improvement of fund performance [17].

## 4.3. Robustness test

This study conducts a robustness test by changing the style drift calculation method and making benchmark adjustments to fund performance. First, to test whether the empirical results will be affected by the construction method of the style drift indicator, this study uses the Manhattan distance between a fund and its performance benchmark to measure the degree of style drift of a fund. $SD\_new_{F,t}$ is similar to the active share measure of Cremers and Petajisto [29]. However, they take 19 reference portfolios, use each reference portfolio in turn as a benchmark, and select the specific portfolio with the lowest activity indicator. In contrast, we use the performance benchmark portfolio disclosed in the fund's prospectus. The specific calculation of this method is:

$$SD\_new_{F,t} = \sum_{j=1}^{N} |w_{F,j,t} - w_{M,j,t}| \tag{5}$$

Where $w_{F,j,t}$ is the weight of industry j in the fund at time t, $w_{M,j,t}$ is the weight of industry j in the fund performance benchmark at time t, and N is the number of industries held by the fund.

Second, excess returns due to industry factors are also momentum factors in the Chinese market. Referring to Mateus et al. [3], this study adjusts the left side of the standard Fama-French model, replacing the risk-free rate with the return of the fund's performance benchmark. This modification aims to control for the impact of industry-average performance on the results.

Table 4 is the robustness test of the regression results of style drift on fund performance. Panel A of Table 4 uses Manhattan distance SD_new to measure fund style drift as a dependent variable to regress original excess returns, CAPM alpha, three-factor alpha, and five-factor alpha.

Panel A shows that after using Manhattan distance to recalculate the degree of fund style drift, the regression coefficients of fund style drift on fund performance are still significantly positive, and they are all significant at the 1% level. Panel B of Table 4 uses the fund performance after benchmark adjustment as the independent variable and performs regression with the style drift index SD. The independent variable after fund performance benchmark adjustment is Rp-Rb, CAPM alpha[b], FF3 alpha[b], FF5 alpha[b]. Panel B shows that after using benchmark-adjusted fund performance as the independent variable, the regression coefficient of fund style drift on fund performance is still significantly positive, and the coefficients of the style drift index SD are all significant at the 1% level. It shows that the empirical results of this study have good robustness.

## 4.4. Endogeneity test

**4.4.1 PSM and DID.** The conclusions of this study may have endogeneity issues, and fund performance may be affected by unobservable factors, such as changes in the market environment, changes in regulations and policies, etc. To examine whether the results suffer from endogeneity, this study employs a difference-in-differences approach to mitigate potential endogeneity issues between fund style drift and performance.

In 2021, China's semiconductor and new energy industries experienced great growth, attracting many funds to buy. As a result, these funds have deviated from the original theme, causing a style drift phenomenon. This phenomenon has attracted the attention of regulatory authorities. In August 2021, the China Securities Journal published a report entitled "Controversy over Style Drift, Fund 'Blind Box' with a Mismatched Name.". The China Securities Regulatory Commission has supervised fund products with deviating styles and required funds to self-examine whether their positions comply with the fund contract. Major custodian banks have also begun to require fund companies to check whether the top 20 holdings of theme funds comply with the contract.

This study uses the CSRC's supervision as an external shock to examine the impact on fund style drift through a difference-in-differences analysis. This study selects high style drift funds and low style drift funds as the experimental group and the control group. Specifically, we take the funds ranked in the top one-third of the style drift degree in the quarter as the experimental group, and the Treat value is 1. The rest are the control group, and the Treat value is 0. In order to reduce the characteristic differences between the experimental group and the control group funds to a certain extent, this study first uses the nearest neighbor matching (1:1) method to perform propensity score matching before the double difference test to obtain fund samples with similar characteristics. The covariates used in propensity score matching are the control variables in Eq (4). Fig 2 shows the kernel density distribution after propensity score matching. It indicates the fitting degree of propensity score matching is high, and the characteristic scores of fund samples after matching tend to be consistent. After matching, 20,000 samples are obtained.

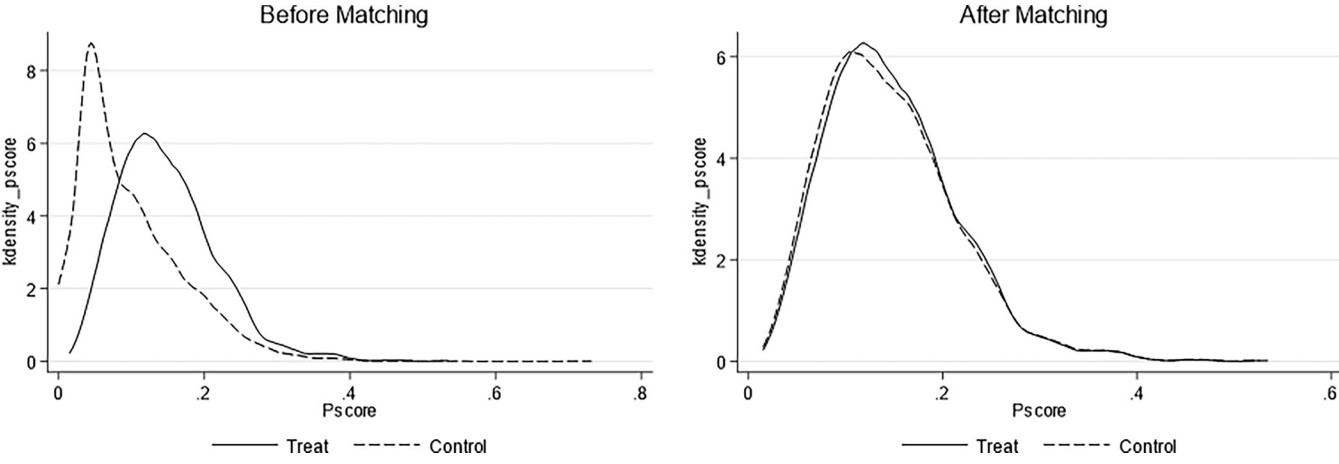

**Fig 2. Propensity score matching kernel density distribution.**

Fig 3 shows the parallel trend test of the fund style drift policy. The x-axis is the policy time point, *current* is the quarter of policy implementation, the period before implementation is *pre_1*, the period after implementation is *post_1* and so on. The y-axis is the fund excess return of the experimental group and the control group. Through the graph, this article can intuitively see that there is no significant difference between the experimental group and the control group at the time point before the policy is implemented. After the policy took effect, the excess returns of the experimental group dropped significantly in the last two periods, indicating that the parallel trend test was passed.

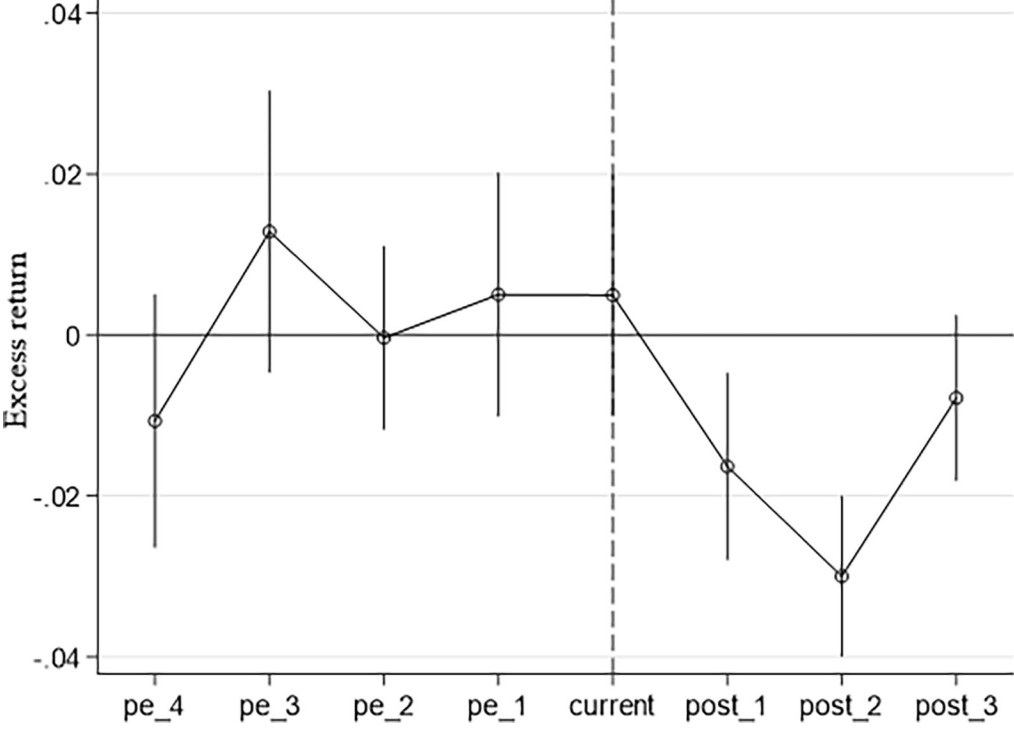

**Fig 3. Trend test for parallel lines.**

**Table 5. DID results.**

|  | (1) | (2) | (3) | (4) |
|---|---|---|---|---|
|  | Rp-Rf | CAPM alpha | FF3 alpha | FF5 alpha |
| Time* Treat | -0.022*** | -0.114*** | -0.111*** | -0.075*** |
|  | (-3.95) | (-6.33) | (-7.17) | (-5.02) |
| Time | -0.080*** | -0.248*** | -0.245*** | -0.158*** |
|  | (-28.84) | (-27.92) | (-32.09) | (-21.35) |
| Treat | 0.010*** | 0.022** | 0.030*** | 0.010 |
|  | (3.07) | (2.14) | (3.37) | (1.52) |
| Controls | Yes | Yes | Yes | Yes |
| Fund FE | No | No | No | No |
| Time FE | No | No | No | No |
| N | 19604 | 19604 | 19604 | 19604 |
| R2 | 0.06 | 0.07 | 0.11 | 0.07 |

***, **, * indicate significance at the 1%, 5%, and 10% significance levels respectively, and the t statistics are in parentheses.

To conduct a DID test on fund style drift, we use the sample after propensity score matching. We add three variables: *Time*, *Treat*, and *Time*\**Treat* to the baseline regression. Other control variables remain consistent with Eq (4). *Time* is a dummy variable denoting the periods after the policy implementation. It is 1 after the CSRC supervision in the third quarter of 2021 and 0 before. *Treat* is a dummy variable, which is 1 for high style drift funds and 0 for low style drift funds. *Time*\**Treat* is the interaction term between *Time* and *Treat*. Its coefficient is the performance difference in high style drift funds, relative to low style drift funds., after the China Securities Regulatory Commission's (CSRC) supervision on style drift.

To mitigate potential endogeneity arising from the impact of style drift on fund performance, this study regresses fund performance on style drift in the DID model. Fund performance in columns (1) to (4) of Table 5 is measured using original excess returns, CAPM alpha, three-factor alpha, and five-factor alpha, respectively. Regression results indicate that the coefficients of *Time*\**Treat* are significantly negative at the 1% significance level across all performance measures. This suggests that, after CSRC's regulatory intervention on style drift, high style drift funds experienced a decrease in style drift relative to low style drift funds. Additionally, the high returns obtained through style drift also decrease.

Taking column (1) as an example, the coefficient of *Time*\**Treat* is -0.022. It indicates that compared to the control group, the CSRC's supervision leads to a significant reduction in the original excess returns of the experimental group, with a policy effect of 0.022. These results demonstrate the reliability of empirical results after mitigating endogeneity.

To test the robustness of the DID test, this study replaces the dummy variable *Treat* with the continuous variable *L_SD*. *L_SD* is the degree of fund style drift in the quarter preceding the supervision implementation. *Time* is a dummy variable, equal to 1 for quarters starting from supervision and 0 for earlier periods. *Time*\**L_SD* is the interaction term between *Time* and *L_SD*, with other control variables remaining consistent with Eq (4).

Table 6 presents the robust results of the DID test. Columns (1) to (4) indicate that the regression coefficient of *Time*\**L_SD* is significantly negative at least at the 10% significance level. These results suggest that, after the CSRC's regulatory intervention on style drift, funds with higher levels of style drift experience a decline in performance. This finding aligns with the conclusions drawn from Table 6.

Table 6. Robustness test of DID: Continuous variables.

| | (1) | (2) | (3) | (4) |
|---|---|---|---|---|
| | Rp-Rf | CAPM alpha | FF3 alpha | FF5 alpha |
| Time* L_SD | -0.128* | -0.767*** | -0.650*** | -0.366* |
| | (-1.66) | (-3.08) | (-3.10) | (-1.81) |
| Time | -0.083*** | -0.253*** | -0.242*** | -0.166*** |
| | (-18.74) | (-17.72) | (-20.20) | (-14.34) |
| L_SD | 0.03 | 0.05 | (0.02) | -0.158* |
| | (0.87) | (0.41) | (-0.22) | (-1.73) |
| Controls | Yes | Yes | Yes | Yes |
| Fund FE | No | No | No | No |
| Time FE | No | No | No | No |
| N | 18949 | 18949 | 18949 | 18949 |
| $R^2$ | 0.06 | 0.06 | 0.12 | 0.07 |

***, **, * indicate significance at the 1%, 5%, and 10% significance levels respectively, and the t statistics are in parentheses.

The results of endogeneity tests demonstrate that the regulatory policy of the CSRC leads to a decrease of 0.022 standard deviations in the original excess returns of funds. The negative impact of policy on style drift appears more in industry theme funds.

### 4.4.2. Instrumental variable

The difficulty in studying fund performance is to eliminate the interference of endogeneity problems. Fund performance may be affected by unobservable factors such as changes in the market environment and changes in regulations and policies. Fund managers may adjust the structure and style of their investment portfolios in order to find the best investment opportunities when facing different market environments, and changes in such market environments will directly affect the performance of the fund.

We use fund managers' voluntary changes (*Change*) as the external impact of portfolio changes and as instrumental variables to reduce the above endogeneity problem. The CSMAR database discloses 11 reasons for fund manager changes, which this article divides into involuntary changes and voluntary changes. Voluntary changes include retirement, expiration of term, change of controlling stake, health reasons, improvement of corporate governance structure, involvement in cases, and termination of agency. Involuntary changes include job transfers, resignations, dismissals, and personal reasons. Since fund managers may be forced to leave due to poor management, involuntary changes in fund managers are often related to fund performance. However, voluntary changes of fund managers, such as retirement, expiration of term, health reasons and other factors have nothing to do with the fund's style drift and performance. Therefore, voluntary changes in fund managers are exogenous to fund performance. New fund managers often sell off their original portfolio stocks to build new investment portfolios. After taking over, new fund managers may adjust the fund's investment portfolio according to their own investment habits. style and experience, which may lead to fund style drift. Therefore, there is a correlation between fund manager changes and fund style drift.

Table 7 shows the regression results of instrumental variables. The first column is the regression result of the endogenous variable SD in the first stage. The instrumental variable Change is significantly positive at the 1% level, which shows that after the fund manager is

**Table 7. Instrumental variable test.**

| | (1) | (2) | (3) | (4) | (5) |
|---|---|---|---|---|---|
| | **SD** | **Rp-Rf** | **CAPM alpha** | **FF3 alpha** | **FF5 alpha** |
| Change | 0.036*** | | | | |
| | (16.05) | | | | |
| SD | | 2.833*** | 11.947*** | 9.456*** | 7.811*** |
| | | (11.89) | (10.65) | (9.67) | (8.52) |
| Controls | Yes | Yes | Yes | Yes | Yes |
| Fund FE | Yes | Yes | Yes | Yes | Yes |
| Time FE | Yes | Yes | Yes | Yes | Yes |
| $N$ | 20547 | 20547 | 20547 | 20547 | 20547 |
| $R^2$ | 0.037 | 0.131 | 0.048 | 0.086 | 0.067 |
| Underidentification test | 255.094*** | | | | |
| Weak identification test | 257.639 | | | | |
| Sargan statistic | 0.000 | | | | |

***, **, * indicate significance at the 1%, 5%, and 10% significance levels respectively, and the t statistics are in parentheses.

replaced, the degree of fund style drift increases. This test passes the over-identification test and the weak instrumental variable test. In columns (2)-(5), the coefficient of SD is still significantly positive, indicating that after using instrumental variables to reduce endogeneity, fund style drift still has a positive relationship with fund performance. This proves that the empirical conclusions of this article are still credible.

## 5. Style drift and fund performance based on fund investment ability

### 5.1. Style drift and fund stock picking ability: Measurement based on return

Fund style drift can occur by adjusting the weights of stocks in different industries and styles in the portfolio. However, fund managers need to possess abilities to forecast market trends, seize investment opportunities, and select among different investment styles. Therefore, the competence of fund managers is crucial for the successful implementation of style drift strategies. Funds with robust stock-picking abilities may have access to more private information about stock values or excel in predicting the directional shifts in industry rotations. Consequently, they might find it easier to generate excess returns through style drift strategies.

This study initially measures whether funds possess investment abilities based on fund returns. If a fund generates excess returns compared to similar stocks, it indicates the fund's stock-picking ability. The Treynor-Mauzy (TM) model examines fund managers' stock-picking abilities by adding a quadratic term of market excess returns to the Capital Asset Pricing Model (CAPM) framework. The model is expressed as follows:

$$R_{it} - R_{ft} = \alpha_{it} + \beta_1(R_{mt} - R_{ft}) + \beta_2(R_{mt} - R_{ft})^2 + \varepsilon_{it} \qquad (6)$$

The constant term $\alpha_{it}$ measures the ability of the fund's portfolio to obtain excess returns, which Treynor and Mauzy attribute to the fund's stock picking ability. This study uses stock picking ability (TM) as the independent variable.

Table 8 reports the regression results of fund style drift on fund stock picking ability to examine the impact mechanism of style drift on fund performance. Column (1) is the fixed

**Table 8. Style drift and fund stock picking ability: TM model.**

|  | (1) | (2) | (3) |
|---|---|---|---|
|  | TM | TM | TM |
| SD | 0.009*** |  |  |
|  | (4.76) |  |  |
| Time*L_SD |  | -0.012** |  |
|  |  | (-2.13) |  |
| L_SD |  | 0.005** |  |
|  |  | (2.00) |  |
| Time* Treat |  |  | -0.002*** |
|  |  |  | (-4.82) |
| Treat |  |  | 0.001*** |
|  |  |  | (5.05) |
| Time |  | -0.006*** | -0.006*** |
|  |  | (-19.55) | (-32.50) |
| Controls | Yes | Yes | Yes |
| Fund FE | Yes | No | No |
| Time FE | Yes | No | No |
| N | 20547 | 18949 | 20547 |
| $R^2$ | 0.490 | 0.080 | 0.090 |

***, **, * indicate significance at the 1%, 5%, and 10% significance levels respectively, and the t statistics are in parentheses.

effect regression of fund style drift on the fund's stock picking ability. The results show that the style drift is positively correlated with the fund's stock picking ability, and the regression coefficient is significant at the 1% statistical level. Specifically, for every one standard deviation increase in the style drift index, the fund's stock picking ability will increase by 0.009 standard deviations. Columns (2) and (3) are the results of the DID test of fund style drift on the fund's stock picking ability.

The results show that the coefficients of both the interaction term *Time*L_SD* and *Time*-Treat* are significantly negative, and the coefficient of *Time*L_SD* is -0.012, which means that for every one standard deviation increase in the fund's style drift index, the fund stock picking ability dropped by 0.012 standard deviations after the CSRC supervision. The coefficient of *Time*Treat* is -0.002. It means the supervision of the China Securities Regulatory Commission caused a significant decline in the stock picking ability of the experimental group, and the policy effect was 0.002. Combining the previous conclusions, this study shows that after the supervision of the China Securities Regulatory Commission, the stock picking ability of style drift funds has declined, which has led to a decrease in fund performance.

Another way to estimate whether a fund has high ability is the Chang and Lewellen model (CL model). The TM model evaluates whether the fund manager can outperform stocks with the same characteristics in selecting stocks. The CL model can analyze the fund manager's ability to choose investment assets when the market trend is falling or rising. The expression of the CL model is as follows:

$$R_{it} - R_{ft} = \alpha_{it} + \beta_1 (R_{mt} - R_{ft}) D_1 + \beta_2 (R_{mt} - R_{ft}) D_2 + \varepsilon_{it} \tag{7}$$

The model divides the market into a long market and a short market, and uses two different values to represent the systemic risks of the two markets, so that the results can better reflect

**Table 9. Style drift and fund stock picking ability: CL model.**

|  | (1) | (2) | (3) |
|---|---|---|---|
|  | CL | CL | CL |
| SD | 0.677*** |  |  |
|  | (5.60) |  |  |
| Time*L_SD |  | -1.874*** |  |
|  |  | (-6.83) |  |
| L_SD |  | 1.178*** |  |
|  |  | (8.99) |  |
| Time* Treat |  |  | -0.067*** |
|  |  |  | (-2.59) |
| Treat |  |  | 0.094*** |
|  |  |  | (4.35) |
| Time |  | -0.960** | 4.892*** |
|  |  | (-2.32) | (11.50) |
| Controls | Yes | Yes | Yes |
| Fund FE | Yes | No | No |
| Time FE | Yes | No | No |
| N | 16946 | 16364 | 16946 |
| $R^2$ | 0.643 | 0.159 | 0.151 |

***, **, * indicate significance at the 1%, 5%, and 10% significance levels respectively, and the t statistics are in parentheses.

the economic significance of the market. When $R_{mt}>R_{ft}$, D1 = 1, D2 = 0. When $R_{mt}<R_{ft}$, D1 = 0, D2 = 1. $\beta_1$ corresponds to the $\beta$ value of the fund in the long market, and $\beta_2$ corresponds to the $\beta$ value of the fund in the short market. The constant term $\alpha_{it}$ measures the ability of the fund's investment portfolio to obtain excess returns, that is, the fund's stock picking ability.

Table 9 uses the CL model to measure the fund's stock picking ability. Column (1) is the fixed effect regression of fund style drift on the fund's stock picking ability. The results show that the style drift is positively correlated with the fund's stock picking ability, and the regression coefficient is significant at the 1% statistical level. Columns (2) and (3) are the results of the DID test of fund style drift on the fund's stock picking ability. The coefficients of the interaction terms *Time*L_SD* and *Time*Treat* are both significantly negative, and the results are consistent with the above.

## 5.2. Style drift and fund stock picking ability: Measurement based on stock holdings

To ensure the robustness of the findings in this study and avoid reliance on methods based on fund returns, we use fund holdings to evaluate a fund's investment strategy. Because merger and acquisition (M&A) transactions of listed companies can be predicted by analyzing data, paying attention to news, and interviewing company executives [30, 31].

Fund managers with stock picking skills are likely to analyze and research to legally find the potential for M&A transactions. Consequently, we examine the shareholdings of funds in the first half year of M&A transactions and identify funds that predict stock M&A investment opportunities as funds with strong capabilities (Ability1). Ability1 is a dummy variable that equals 1 if a fund predicts at least one stock acquisition, otherwise, it is 0.

We add the fund investment ability variable and the interaction term between fund investment ability and style drift as dependent variables into Eq (8). The other variables remain unchanged. The empirical model is outlined below:

$$\alpha_{it} = \beta_0 + \beta_1 SD_{it-1} + \beta_2 Ability1_{it-1} + \beta_3 SD_{it-1}*Ability1_{it-1} + \beta_4 Controls_{it-1} + Quarter_t + Fund_i + \varepsilon_{it} \quad (8)$$

Table 10 tests the importance of fund investment ability, that is, whether funds with strong ability can achieve better performance through style drift. The results in columns (1)-(4) of Table 10 show that the interaction term ($SD_{it-1}*Ability1_{it-1}$) of the fund's investment ability and style drift has a significantly positive impact on the fund's original excess return, CAPM alpha, three-factor alpha and five-factor alpha, and significant at least at the 5% level.

Taking column (1) as an example, the coefficient of the interaction term between SD and fund investment ability is 0.080, which means that when the style drift SD increases by one standard deviation, the original excess return of a fund with strong investment ability will increase 0.080 standard deviations more than that of a fund with weak investment ability. This

**Table 10. Fund investment ability and fund performance: Whether to predict M&A stocks.**

| | (1) | (2) | (3) | (4) |
|---|---|---|---|---|
| | Rp-Rf | CAPM alpha | FF3 alpha | FF5 alpha |
| SD | 0.066*** | 0.139 | 0.112 | 0.067 |
| | (2.58) | (1.05) | (0.91) | (0.55) |
| Ability1 | -0.004** | 0.001 | -0.006 | -0.012 |
| | (-2.38) | (0.10) | (-0.80) | (-1.47) |
| SD* Ability1 | 0.080*** | 0.395*** | 0.279** | 0.408*** |
| | (2.76) | (2.64) | (2.02) | (2.97) |
| Size | 0.002** | 0.009** | 0.016*** | 0.009*** |
| | (2.38) | (2.42) | (4.42) | (2.58) |
| Fundage | -0.020*** | -0.036** | -0.027* | -0.019 |
| | (-6.24) | (-2.17) | (-1.76) | (-1.24) |
| Conc | -0.000 | 0.001 | -0.034*** | -0.027*** |
| | (-0.27) | (0.18) | (-5.37) | (-4.26) |
| Change | -0.005*** | -0.033*** | -0.024*** | -0.023*** |
| | (-3.70) | (-4.56) | (-3.64) | (-3.41) |
| Gender | -0.012** | -0.053** | -0.041* | -0.045** |
| | (-2.50) | (-2.16) | (-1.83) | (-1.97) |
| Edu | -0.000 | -0.009 | -0.012 | -0.006 |
| | (-0.01) | (-0.35) | (-0.51) | (-0.25) |
| Work | -0.001* | -0.004 | 0.000 | 0.003 |
| | (-1.85) | (-1.36) | (0.12) | (0.96) |
| Serv | -0.000 | 0.002 | -0.000 | -0.002 |
| | (-0.00) | (0.62) | (-0.00) | (-0.58) |
| Age | 0.001 | 0.005 | -0.003 | 0.003 |
| | (0.57) | (0.80) | (-0.51) | (0.48) |
| Fund FE | Yes | Yes | Yes | Yes |
| Time FE | Yes | Yes | Yes | Yes |
| N | 20547 | 20547 | 20547 | 20547 |
| $R^2$ | 0.774 | 0.395 | 0.289 | 0.223 |

***, **, * indicate significance at the 1%, 5%, and 10% significance levels respectively, and the t statistics are in parentheses.

**Table 11. Fund investment ability and fund performance: Number of predicting M&A stocks.**

|  | (1) | (2) | (3) | (4) |
|---|---|---|---|---|
|  | Rp-Rf | CAPM alpha | FF3 alpha | FF5 alpha |
| SD | 0.081*** | 0.229* | 0.167 | 0.170 |
|  | (3.43) | (1.90) | (1.50) | (1.53) |
| Ability2 | -0.001** | 0.002 | -0.004 | -0.005* |
|  | (-2.05) | (0.59) | (-1.41) | (-1.78) |
| SD* Ability2 | 0.028** | 0.121** | 0.090* | 0.114** |
|  | (2.44) | (2.07) | (1.68) | (2.13) |
| Size | 0.002** | 0.010** | 0.015*** | 0.009** |
|  | (2.38) | (2.47) | (4.32) | (2.46) |
| Fundage | -0.020*** | -0.036** | -0.027* | -0.019 |
|  | (-6.23) | (-2.20) | (-1.78) | (-1.25) |
| Conc | -0.000 | 0.001 | -0.033*** | -0.026*** |
|  | (-0.31) | (0.15) | (-5.34) | (-4.24) |
| Change | -0.005*** | -0.034*** | -0.024*** | -0.023*** |
|  | (-3.73) | (-4.73) | (-3.64) | (-3.42) |
| Gender | -0.012** | -0.054** | -0.041* | -0.044* |
|  | (-2.48) | (-2.21) | (-1.81) | (-1.95) |
| Edu | 0.000 | -0.007 | -0.011 | -0.005 |
|  | (0.01) | (-0.27) | (-0.48) | (-0.21) |
| Work | -0.001* | -0.004 | 0.000 | 0.003 |
|  | (-1.84) | (-1.39) | (0.12) | (0.96) |
| Serv | -0.000 | 0.003 | -0.000 | -0.002 |
|  | (-0.02) | (0.68) | (-0.03) | (-0.62) |
| Age | 0.001 | 0.006 | -0.003 | 0.003 |
|  | (0.60) | (0.87) | (-0.49) | (0.49) |
| Fund FE | Yes | Yes | Yes | Yes |
| Time FE | Yes | Yes | Yes | Yes |
| N | 20547 | 20547 | 20547 | 20547 |
| $R^2$ | 0.774 | 0.395 | 0.289 | 0.223 |

***, **, * indicate significance at the 1%, 5%, and 10% significance levels respectively, and the t statistics are in parentheses.

shows that the fund's investment ability promotes the performance improvement of style drift funds.

To test the robustness of the conclusion, this study also uses another alternative index (*Ability*2) to measure the fund's investment ability. *Ability*2 is a continuous variable, which is the number of all M&A investment opportunities that the fund has predicted through information. Table 11 uses *Ability*2 as a proxy for fund investment ability. Consistent with the findings in Table 10, the results show that when a fund has a higher ability in style drift, it is more conducive to improving fund performance.

## 6. Style drift and fund performance of chasing market trends

This study calculates the fund's increase ratio in the current period for the top ten industries with the highest growth rate in the previous quarter, as a measure of the fund's degree of chasing hot trends (*Chasing1*). The more the fund increases its holdings in the popular industries of the previous quarter, the greater the fund's degree of chasing hot trends. We introduce the fund chasing hot trends variable and the interaction term between chasing hot trends and style

**Table 12. Fund chasing market trend and fund performance: Shareholding increase ratio of the top ten.**

|  | (1) | (2) | (3) | (4) |
|---|---|---|---|---|
|  | **Rp-Rf** | **CAPM alpha** | **FF3 alpha** | **FF5 alpha** |
| SD | 0.155*** | 0.587*** | 0.467*** | 0.530*** |
|  | (7.26) | (5.34) | (4.59) | (5.24) |
| Chasing1 | 0.002 | -0.000 | 0.110** | -0.069 |
|  | (0.18) | (-0.01) | (2.41) | (-1.53) |
| SD*Chasing1 | -0.386*** | -1.980*** | -2.213*** | -1.738*** |
|  | (-3.51) | (-3.50) | (-4.23) | (-3.34) |
| Size | 0.002** | 0.008** | 0.015*** | 0.008** |
|  | (2.29) | (2.08) | (4.35) | (2.31) |
| Fundage | -0.020*** | -0.036** | -0.027* | -0.019 |
|  | (-6.23) | (-2.20) | (-1.77) | (-1.25) |
| Conc | -0.000 | 0.003 | -0.034*** | -0.025*** |
|  | (-0.18) | (0.40) | (-5.39) | (-3.99) |
| Change | -0.005*** | -0.033*** | -0.024*** | -0.023*** |
|  | (-3.74) | (-4.55) | (-3.66) | (-3.42) |
| Gender | -0.012** | -0.053** | -0.041* | -0.045** |
|  | (-2.52) | (-2.16) | (-1.82) | (-2.00) |
| Edu | 0.000 | -0.007 | -0.011 | -0.004 |
|  | (0.01) | (-0.27) | (-0.49) | (-0.19) |
| Work | -0.001* | -0.004 | 0.000 | 0.003 |
|  | (-1.83) | (-1.37) | (0.10) | (0.98) |
| Serv | -0.000 | 0.002 | -0.000 | -0.002 |
|  | (-0.05) | (0.53) | (-0.02) | (-0.68) |
| Age | 0.001 | 0.005 | -0.003 | 0.003 |
|  | (0.53) | (0.77) | (-0.55) | (0.45) |
| Fund FE | Yes | Yes | Yes | Yes |
| Time FE | Yes | Yes | Yes | Yes |
| $N$ | 20547 | 20547 | 20547 | 20547 |
| $R^2$ | 0.774 | 0.396 | 0.289 | 0.225 |

***, **, * indicate significance at the 1%, 5%, and 10% significance levels respectively, and the t statistics are in parentheses.

drift into Eq (9) as independent variables while keeping other variables constant. Dependent variables are the fund's original excess return, CAPM alpha, three-factor alpha, and five-factor alpha. The empirical model is constructed as follows:

$$\alpha_{it} = \beta_0 + \beta_1 SD_{it-1} + \beta_2 Chasing1_{it-1} + \beta_3 SD_{it-1}*Chasing1_{it-1} + \beta_4 Controls_{it-1} + Quarter_t + Fund_i + \varepsilon_{it} \quad (9)$$

Table 12 examines the impact of the fund's degree of chasing hot trends on its performance. It shows the interactive impact of chasing hot trends and style drift on fund performance. The results in columns (1)-(4) of Table 12 show that the interaction term ($SD_{it}*Chasing1_{it}$) all are significant at the 1% level.

Taking column (1) as an example, the coefficient of $SD_{it-1}*Chasing1_{it-1}$ is -0.386, which means that for every one standard deviation increase in the chasing hot trends, the fund's original excess return will decrease by 0.386 standard deviations. This shows that the effect of fund style drift on improving fund performance doesn't come from chasing popular industries. Funds cannot improve their performance by chasing market trends. This may be due to the

**Table 13. Fund chasing market trend and fund performance: Shareholding increase ratio of the top five.**

| | (1) | (2) | (3) | (4) |
|---|---|---|---|---|
| | **Rp-Rf** | **CAPM alpha** | **FF3 alpha** | **FF5 alpha** |
| SD | 0.146*** | 0.518*** | 0.421*** | 0.482*** |
| | (7.19) | (4.95) | (4.34) | (5.00) |
| Chasing2 | -0.067*** | -0.288*** | -0.046 | -0.072 |
| | (-4.69) | (-3.94) | (-0.67) | (-1.07) |
| SD*Chasing2 | -0.322** | -1.547* | -2.550*** | -2.998*** |
| | (-1.97) | (-1.84) | (-3.28) | (-3.88) |
| Size | 0.002** | 0.008** | 0.015*** | 0.008** |
| | (2.16) | (1.98) | (4.25) | (2.34) |
| Fundage | -0.020*** | -0.037** | -0.027* | -0.019 |
| | (-6.27) | (-2.22) | (-1.76) | (-1.24) |
| Conc | 0.000 | 0.004 | -0.033*** | -0.025*** |
| | (0.03) | (0.55) | (-5.23) | (-4.09) |
| Change | -0.005*** | -0.032*** | -0.024*** | -0.023*** |
| | (-3.71) | (-4.53) | (-3.63) | (-3.41) |
| Gender | -0.012*** | -0.054** | -0.042* | -0.046** |
| | (-2.58) | (-2.21) | (-1.86) | (-2.02) |
| Edu | -0.000 | -0.008 | -0.011 | -0.005 |
| | (-0.02) | (-0.30) | (-0.49) | (-0.23) |
| Work | -0.001* | -0.004 | 0.001 | 0.003 |
| | (-1.66) | (-1.22) | (0.21) | (1.08) |
| Serv | -0.000 | 0.002 | -0.000 | -0.003 |
| | (-0.20) | (0.40) | (-0.13) | (-0.74) |
| Age | 0.001 | 0.005 | -0.004 | 0.002 |
| | (0.47) | (0.72) | (-0.59) | (0.37) |
| Fund FE | Yes | Yes | Yes | Yes |
| Time FE | Yes | Yes | Yes | Yes |
| N | 20547 | 20547 | 20547 | 20547 |
| $R^2$ | 0.775 | 0.398 | 0.290 | 0.226 |

***, **, * indicate significance at the 1%, 5%, and 10% significance levels respectively, and the t statistics are in parentheses.

increase in non-systemic risks, rising transaction costs, and lack of information caused by chasing market trends.

To reduce the impact of the chasing market trends index construction method and increase the robustness of the conclusion, this study also uses another alternative indicator (*Chasing*2) to measure the degree of funds chasing market trends. *Chasing*2 is the fund's increase ratio for the top five industries with the highest growth rate in the previous quarter. Table 13 shows the regression of fund performance on fund style drift and the degree of chasing market trends, using *Chasing*2 as a proxy variable for the degree of chasing market trends. Consistent with the findings in Table 12, the results show that fund style drift does not improve fund performance by chasing popular industries.

## 7. Conclusion

This study examines two distinct types of fund style drift. In 2022, the China Securities Regulatory Commission (CSRC) issued the "Opinions on Accelerating the High-Quality Development of the Public Fund Industry", which implemented regulations on fund style drift

behavior. However, research on fund style drift has found that it enhances fund performance. This study further differentiates between different types of style drift, to study this issue that is both academically debated and conflicts with practical policy.

The results show that, overall, fund style drift significantly improves fund performance. We further distinguish between style drift driven by investment ability and style drift driven by chasing market trends.We measure the fund's ability from the perspective of fund returns and shareholdings. We regress fund performance on the fund's investment ability, style drift, and interaction terms. The results show that fund style drift with higher ability can enhance fund performance. Additionally, the study measures the fund chasing market trends by calculating the increase ratio of the top ten industries with the highest growth rate. The study finds a significant negative correlation between chasing hot trends and performance.

The empirical results suggest that whether funds should adopt a drifting strategy depends on the type of drift. Funds with stock picking abilities can benefit from style drift, enhancing their performance. On the contrary, style drift induced by chasing hot trends may damage performance. Funds with stock picking abilities can identify market opportunities, obtain more private information, and choose between different investment styles. In contrast, chasing hot trends through style drift is a short-sighted behavior, resulting in increased fund transaction costs and non-systematic risks.

The conclusions also hold practical significance for the capital market. As more funds choose style drift to enhance their performance, securities regulatory authorities should take measures to protect investors' interests and promote the high-quality development of the public fund industry. However, regulatory authorities should not adopt a uniform approach to regulate fund style drift but rather differentiate based on the type of drift. regulatory authorities should apply mandatory regulatory measures to fund style drift driven by chasing hot trends, while market-oriented approaches to funds style drift with stock picking abilities. For fund investors, it is crucial not to blindly pursue high returns and popular funds and enhance their ability to assess fund returns. This study also emphasizes the importance of investors' education and risk awareness. Investors should deepen their understanding of different types of funds, and focus not only on high returns but also consider fund investment strategies and risk factors. Investment decisions should be aimed at achieving long-term stable returns rather than being influenced by short-term market trends.

## 8. Discussion

The impact of style drift on fund performance remains a contentious issue in the literature. Some studies argue that style drift reflects the active management ability of fund managers. By adjusting investment styles to adapt to market changes, fund managers can mitigate risks or seize investment opportunities, thereby generating excess returns [20]. This finding is consistent with our baseline regression results. However, other research suggests that style drift weakens fund performance [17]. Our findings differ from the existing literature in several key aspects.

First, this study adopts a distinct perspective. Previous studies primarily measured style drift based on indicators such as stock size, book-to-market ratio, and momentum, which focus on fundamental characteristics [18]. In contrast, our study examines style drift from the perspective of industry allocation. Second, our approach differs from that of Liu et al. [4], who measured style drift through industry allocation differences over different periods, making it more suited for analyzing historical trends within individual funds. In contrast, our study measures style drift by comparing the industry allocation differences between the fund and its performance benchmark at the same time period, providing a cross-sectional comparison that is

more suitable for analyzing multiple funds, allowing us to reveal whether funds deviate from their investment objectives. Additionally, Liu et al. [4] used annual data, whereas this study uses quarterly data. Since changes in a fund's investment style do not necessarily occur only at the year-end, quarterly data provides a reflection of style adjustments and performance variations.

Furthermore, previous studies have overlooked the behavioral differences in style drift. The impact of style drift on performance may depend on the fund manager's motivation. Differences in motivation can lead to varying effects on performance [15, 16]. This study distinguishes between investment ability-driven drift and trend-chasing drift. Research indicates that fund managers with strong stock-picking or timing abilities tend to enhance performance when adjusting their portfolios [14, 16], which aligns with our findings: funds with strong stock-picking abilities achieve higher returns when engaging in style drift. On the other hand, trend-chasing style drift tends to harm fund performance because frequent adjustments increase transaction costs, misjudgments, and decision-making risks [17], ultimately leading to underperformance [18]. By analyzing the effects of different types of style drift, this study not only explains the discrepancies in the literature but also reveals the complexity of how style drift affects fund performance, offering a more comprehensive theoretical and empirical basis for fund research.

This study does have certain limitations that should be addressed in future research. First, while we measure style drift through changes in industry allocation ratios, this approach effectively captures the intensity and frequency of drift but does not fully analyze its direction. Future studies could build on this method by incorporating measures of the direction of drift to identify which industries funds are likely to tilt towards under different market conditions. This would enhance the accuracy of assessing the impact of industry performance on fund returns and allow for a better understanding of fund preferences across economic cycles, providing more actionable insights for fund management and risk control.

Moreover, while this study investigates the impact of style drift on performance, it does not examine how this impact propagates within the fund network and its potential effect on overall market performance. The fund market is a highly interconnected network, where the style drift and performance fluctuations of individual funds can transmit to other funds through overlapping holdings or market sentiment, potentially leading to market-wide volatility. Future research could adopt complex network analysis to explore how style drift propagates across a network of funds, especially during periods of market turbulence. In such times, style drift in some funds could trigger a cascading effect, influencing overall market stability. By identifying key nodes within the fund network, future research could provide regulators with targeted early-warning systems, thereby enhancing the identification of systemic risks and improving market stability.

## Author Contributions

**Methodology:** Yaozhi Chen.

**Software:** Yaozhi Chen.

**Supervision:** Honghong Wei.

**Visualization:** Honghong Wei.

**Writing – original draft:** Yaozhi Chen.

**Writing – review & editing:** Honghong Wei.

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
