## [Decision Letter · Decision Letter 0]

18 Oct 2024

PONE-D-24-36296Fund Style Drift and Fund Performance: Evidence from ChinaPLOS ONE

Dear Dr. Chen,

Thank you for submitting your manuscript to PLOS ONE. After careful consideration, we feel that it has merit but does not fully meet PLOS ONE’s publication criteria as it currently stands. Therefore, we invite you to submit a revised version of the manuscript that addresses the points raised during the review process.

**ACADEMIC EDITOR: **

The manuscript explores an interesting research topic, but it could be further enhanced by incorporating the reviewers' suggestions. Additionally, the abstract would benefit from more detail to better reflect the entire study. Including a brief summary of the methodology used would be particularly helpful in providing a clearer overview.

We look forward to receiving your revised manuscript.

Kind regards,

Kingstone Nyakurukwa

Academic Editor

PLOS ONE

Journal Requirements:

2. Please note that your Data Availability Statement is currently missing [the repository name and/or the DOI/accession number of each dataset OR a direct link to access each database]. If your manuscript is accepted for publication, you will be asked to provide these details on a very short timeline. We therefore suggest that you provide this information now, though we will not hold up the peer review process if you are unable.

Reviewers' comments:

Reviewer's Responses to Questions

**Comments to the Author**

1. Is the manuscript technically sound, and do the data support the conclusions?

Reviewer #1: Yes

Reviewer #2: Partly

2. Has the statistical analysis been performed appropriately and rigorously? 

Reviewer #1: Yes

Reviewer #2: Yes

3. Have the authors made all data underlying the findings in their manuscript fully available?

Reviewer #1: No

Reviewer #2: Yes

4. Is the manuscript presented in an intelligible fashion and written in standard English?

Reviewer #1: No

Reviewer #2: Yes

5. Review Comments to the Author

Reviewer #1: Dear author(s),

I have read your paper and found it very interesting, but still, I have found some major issues and fundamental flaws which needs to be addressed before publication.

1- Abstract: Abstract is the executive summary of all of your paper, however, you have not provided details about methodology and implication.

2- Introduction: In this section you are providing a brief introduction about the issue your investigating in global as well as target context with the help of data, tables, reports, numbers, figures, and justifying your statements by references. In your case, you have mixed your Introduction and Literature review section. It is very difficult to continue the flow and difficult to understand. You have not justified your statements i.e. page 11 para 2 and page 12 para 3,4. Please provided proper justification (citations) for all of your important statements.

3- Literature Review: It is one of the most important section, in which you are providing what are the "known and unknown" about your topic by providing the latest references to justify your study. In your case, most of the reference are outdated, which shows that nothing has been done in this area or you missed some important information. You have also interpreted some of your results in LR section on page 11 &12.

Please rewrite you LR section and discuss the variables and their relationships with respect to previous studies, provide theoretical support for each of the relationship and come-up with what has been unknown or what needs to be investigated and develop your hypothesis to be tested.

4- Methodology: In this section you are discussing your sampling procedure, variable measurement, estimation model, hypothesis testing and justifying each one by providing references.

5- Findings/Results: On page 17, Figure1 Panel B, at the bottom of the table, you mentioned that "t-statistics is in parenthesis" however, there is no parenthesis in the table. Panel B of Table 1, most of the interpretation and of the significance are wrong, i.e. CAMP alpha, Size, Fundage etc.

6- Discussion: In this section, you have to compare your result with previous studies and provided the justification for the consistency or differences. In your paper, you have not provided the discussion.

7- Writing: You have to proofread your paper paper as i have noted number of grammatical mistakes.

Overall; The overall papers is looks like a copy paste from a thesis. You have to re-write the paper with proper format; includes Introduction, LR, Methodology, Results/Findings, Discussion and Conclusion.

Thank you

Reviewer #2: Title: Fund Style Drift and Fund Performance: Evidence from China

Abstract: The abstract looks too normal. It must be a little innovative to catch the readers' attention. The abstract must include more outcomes from the study, as well as policy implications and recommendations. It must also identify the years, scope, and method applied in the research.

Introduction

• The introduction must be reorganized properly. It contains too many smaller paragraphs, which leads to inconsistency and a lack of flow. Most of these paragraphs must be combined.

• The introduction lacks adequate background information as well as statistics to support the idea and readings

• It would be beneficial to clearly state the primary research objectives and focus of your study in the introduction. Consider restructuring your paper to provide a clearer organization of ideas.

• Elaborate on key concepts such as the relationship between the main variables. Provide more detailed explanations and examples to help readers grasp the concepts. Consider integrating recent research to support your research problem and objectives.

• The novelty including the main contributions of the study is missing. The introduction which serves as the bedrock of the study is not what I am expecting to see in a top journal like this.

Literature

• This section needs to benefit from a theoretical and empirical literature review to buttress the background of the study. These two sections must be separated in the study

• A literature gap must be developed in the study to capture the differences between the study and previous literature

• I will recommend that a clear hypothesis be developed in this section.

• The authors must end this part by concluding their contribution to the extant literature.

Data

• This section must clearly provide some justification for the use of the variables employed in the study.

• More justification is needed on the time span used as well as the countries selected. Avoid stating lack of data as reasons for the selection of time and countries.

• A table must be provided in this section and must include variable definition, data source, years and expected signs

• This section must provide some justification for the use of the variables employed in the study.

Methods

• This section must clearly provide some justification for the use of the variables employed in the study.

• More justification is needed on the time span used as well as the countries selected. Avoid stating lack of data as reasons for the selection of time and countries.

• Why did the study dealt with SSA countries?

Equations and methods

• A detailed equations and methodological stages must be stated.

• All the methods employed in the study should be grouped under one heading. The importance of these methods must be explained to capture their relevance and their applications.

Results

• The discussion of results should emphasize the economic logic and intuition behind the results.

• I will recommend that the author links the discussion with other studies

• Don’t forget to explicitly explain the results.

Conclusion and Policy Implication

• In the conclusions section, a good one must be given.

• The implications and recommendations must be related to the results of the study

• More cutting-edge limitations and direction for future studies must be stated

6. PLOS authors have the option to publish the peer review history of their article (what does this mean?). If published, this will include your full peer review and any attached files.

Reviewer #1: **Yes: **Maeenuddin

Reviewer #2: **Yes: **Michael Appiah

---

## [Author Response · Author response to Decision Letter 0]

1 Dec 2024

Reviewer #1 Response

Reviewer #1: 

Dear author(s),

I have read your paper and found it very interesting, but still, I have found some major issues and fundamental flaws which needs to be addressed before publication.

1- Abstract: Abstract is the executive summary of all of your paper, however, you have not provided details about methodology and implication.

Response: Thank you for pointing out this issue. We have revised the abstract to include more specific details regarding the methodology and implications. In particular, we describe the bidirectional fixed effects model used for data analysis and introduce a new measurement method for fund style drift based on industry allocation. Additionally, we have elaborated on the practical implications of our findings for fund managers and regulators. We hope these revisions address your concerns comprehensively.

2- Introduction: In this section you are providing a brief introduction about the issue your investigating in global as well as target context with the help of data, tables, reports, numbers, figures, and justifying your statements by references. In your case, you have mixed your Introduction and Literature review section. It is very difficult to continue the flow and difficult to understand. You have not justified your statements i.e. page 11 para 2 and page 12 para 3,4. Please provided proper justification (citations) for all of your important statements.

Response: Thank you for your valuable feedback. We have carefully revised the introduction, incorporating data, numbers, and reports in the research background to explain better the issues we are investigating. We have provided relevant data on the development scale of actively managed equity funds in China and the prevalence of style drift. Additionally, we have included references to the concept of style drift and its correlation, citing key literature on the subject. Furthermore, we have restructured the introduction to clearly distinguish it from the literature review, addressing your concerns regarding the flow and structure of this section. We now provide a more concise introduction to the research questions, followed by a dedicated literature review that discusses related studies in detail. This reorganization improves the clarity and logical flow of the paper. For the statements in the second paragraph on page 11 and the third and fourth paragraphs on page 12, we have added justifications for the relevant statements (with citations).

3- Literature Review: It is one of the most important section, in which you are providing what are the "known and unknown" about your topic by providing the latest references to justify your study. In your case, most of the reference are outdated, which shows that nothing has been done in this area or you missed some important information. You have also interpreted some of your results in LR section on page 11 &12.

Please rewrite you LR section and discuss the variables and their relationships with respect to previous studies, provide theoretical support for each of the relationship and come-up with what has been unknown or what needs to be investigated and develop your hypothesis to be tested.

Response: Thank you for your valuable feedback. We have carefully rewritten the literature review section to address the concerns you raised. Specifically, we have made the following revisions:

1. Updated and Expanded the Literature Review: We acknowledge the concern that many of the references in the original version may be outdated. To address this, we have incorporated more recent studies to provide a more comprehensive understanding of the research landscape. These include studies from 2021 and 2024 (Li and Jin, 2021; Koenig and Burghof, 2022; Yi et al., 2024; Liu and Yi, 2024; Zhang and Lv, 2024), which discuss the factors influencing fund style drift and its impact on fund performance. By integrating these newer sources, we aim to reflect the latest developments in the field and offer a clearer understanding of both the known and unknown aspects of the topic.

2. Clarified the Relationships Between Variables: We have discussed each key variable in the study about prior research, providing theoretical support for each relationship and formulating corresponding research hypotheses. We explore both the positive and negative impacts of style drift on fund performance, as discussed in previous studies (e.g., Wermers, 2012; Andreu et al., 2019), and address the reasons behind the conflicting findings in the literature (Huang, 2011; Zhang et al., 2024; Yi et al., 2024). Additionally, in the final paragraph of the literature review, we summarize the relationship between our study and previous research, highlighting areas that warrant further investigation.

3. Identified Knowledge Gaps and Formulated Hypotheses: In response to your comment on addressing the "unknowns" or "gaps" in research, we identified several key gaps in the literature. Specifically, the relationship between industry-style drift and fund performance has not been sufficiently explored, particularly in the context of China. Most existing studies focus on style drift based on market capitalization or growth/value characteristics but do not delve into how industry allocation influences performance. We propose a more targeted measurement of style drift, focusing on industry-level deviations, which have not been adequately addressed in the literature. We also highlight the lack of research on the motivations behind style drift, especially distinguishing between drift driven by investment ability and trend-chasing drift, and how these two types of drift differentially impact performance.

4. Revised Hypotheses: Based on the latest literature and identified gaps, we have added and clarified our hypotheses. We now explicitly test the relationship between different types of style drift and fund performance.

4- Methodology: In this section you are discussing your sampling procedure, variable measurement, estimation model, hypothesis testing and justifying each one by providing references.

Response: Thank you for your feedback. We have revised the methodology section to provide clearer explanations and justifications for each part. Below are the key changes we made:

1. We have specified the data sources from the WIND and RESSET databases and explained the rationale for selecting Chinese open-end funds from 2007 to 2022 as the research sample. The Chinese fund market is highly influenced by policies, and regulatory bodies such as the China Securities Regulatory Commission (CSRC) have implemented a series of policy documents to limit style drift and regulate market behavior. This policy intervention provides a unique context for studying fund style drift and its impact. The decision to start the study in 2007 was made for two reasons: first, the number of Chinese open-end equity funds was relatively small before 2007, and second, fund information disclosure was incomplete before this time.

2. We have also added references for each variable used in the variable measurement section, which can be found on page 10. Furthermore, we have clarified that the empirical model includes both fund-fixed effects and time-fixed effects to account for unobserved heterogeneity.

5- Findings/Results: On page 17, Figure1 Panel B, at the bottom of the table, you mentioned that "t-statistics is in parenthesis" however, there is no parenthesis in the table. Panel B of Table 1, most of the interpretation and of the significance are wrong, i.e. CAMP alpha, Size, Fundage etc.

Response: Thank you for your observation. We have corrected the labeling at the bottom of Table 1 regarding the “t-statistics in parentheses” to ensure that the table labels are consistent with the actual table content. Additionally, we have revised the interpretation of the results, particularly regarding the significance of CAPM alpha, Size, Fund Age, and other variables. The difference between the average original excess return and CAPM alpha is significant at the 1% level, and the difference between three-factor alpha and five-factor alpha is significant at the 5% level.

6- Discussion: In this section, you have to compare your result with previous studies and provided the justification for the consistency or differences. In your paper, you have not provided the discussion.

Response: Thank you for your feedback. We have added a discussion section to address your comments. In the updated version, we compare our findings with previous research in Chapter 8 and provide explanations for observed consistencies or discrepancies, as well as outline the limitations of our study and potential directions for future research. Specifically:

1. Our study differs from previous literature in several ways. First, we examine style drift from an industry allocation perspective, whereas previous studies often focused on fundamental characteristics such as stock size or book-to-market ratios. Second, we measure style drift using benchmark industry allocation differences, which allow for cross-fund comparisons and help identify whether a fund deviates from its investment objective. Furthermore, our use of quarterly data enables a more timely reflection of style adjustments, whereas previous studies predominantly relied on annual data.

2. Additionally, we further explore the behavioral differences in style drift, distinguishing between investment ability-driven drift and trend-chasing drift. Our findings suggest that fund managers with strong stock-picking or market-timing abilities can enhance fund performance through style drift while trend-chasing drift tends to lead to performance deterioration due to increased transaction costs and decision risks. By analyzing the effects of different types of style drift, this study not only explains the discrepancies in existing research but also reveals the complexity of style drift's impact on performance.

However, there are still some limitations in this study. Future research could further improve the measurement of style drift. Although we measure style drift by changes in industry allocation proportions, which effectively capture the intensity and frequency of drift, we have not precisely analyzed the direction of the drift. Future studies could incorporate measures of drift direction to more accurately assess the impact of industry performance on funds. Moreover, this study does not explore the transmission mechanism of style drift within fund networks or its broader market impact. The fund market is a highly interconnected network, and future research could use complex network analysis to investigate how style drift spreads across funds and its potential impact on the overall market, thereby providing regulators with more targeted early warning mechanisms.

7- Writing: You have to proofread your paper paper as i have noted number of grammatical mistakes.

Response: Thank you for pointing this out. We have carefully proofread the entire paper and corrected the identified grammatical errors.

Overall; The overall papers is looks like a copy paste from a thesis. You have to re-write the paper with proper format; includes Introduction, LR, Methodology, Results/Findings, Discussion and Conclusion. Thank you

Response: Thank you for your valuable feedback. We have carefully revised the structure of the paper to ensure it adheres to the correct academic format. The paper is now divided into distinct sections: Introduction, Literature Review, Methodology, Results/Findings, Discussion, and Conclusion, in line with standard academic structure. We have also ensured that each section is written coherently, maintaining a logical flow from one section to the next, with appropriate transitions and a more formal academic tone.

Reviewer #2 Response

Reviewer #2: Title: Fund Style Drift and Fund Performance: Evidence from China

Abstract: The abstract looks too normal. It must be a little innovative to catch the readers' attention. The abstract must include more outcomes from the study, as well as policy implications and recommendations. It must also identify the years, scope, and method applied in the research.

Introduction

• The introduction must be reorganized properly. It contains too many smaller paragraphs, which leads to inconsistency and a lack of flow. Most of these paragraphs must be combined.

Response: Thank you for your constructive feedback. We have made the following revisions based on your suggestions:

In the abstract, we introduced the new methodology used in this paper to make it more engaging and informative. We also highlighted the key findings of the study, such as the relationship between fund style drift and fund performance, along with policy recommendations for fund managers and regulators. Additionally, we clarified the research years and scope of the sample used in the study, and elaborated on the methodology, including the regression model based on industry allocation drift and various risk-adjusted return models. We have reorganized the introduction to improve its flow and coherence. Smaller paragraphs have been merged to create a more cohesive narrative.

• The introduction lacks adequate background information as well as statistics to support the idea and readings

Response: Thank you for your valuable comments. We have revised the introduction to include more comprehensive background information and supporting statistics. Specifically, we have added relevant data to highlight the growth and significance of the Chinese fund market (e.g., the number of actively managed equity funds and total assets). We also incorporated previous research and statistical data to emphasize the prevalence of the style drift phenomenon. These additions provide a stronger foundation for the study and better support the research questions and objectives.

• It would be beneficial to clearly state the primary research objectives and focus of your study in the introduction. Consider restructuring your paper to provide a clearer organization of ideas.

Response: Thank you for your valuable suggestions. We have revised the introduction to clearly state the main research objectives and focus. The updated introduction now provides a clear overview of the purpose of studying fund style drift and its impact on fund performance within the context of the Chinese market. Additionally, we have reorganized the introduction to improve the organization of ideas, ensuring a clearer flow from the background to the research questions and objectives.

• Elaborate on key concepts such as the relationship between the main variables. Provide more detailed explanations and examples to help readers grasp the concepts. Consider integrating recent research to support your research problem and objectives.

Response: Thank you for your insightful comments. We have revised the introduction to provide a more detailed explanation of key concepts, particularly the relationship between main variables such as fund style drift and fund performance. We now explain in detail how style drift can either enhance or diminish fund performance, depending on underlying motivations and market conditions. Additionally, we have provided examples to illustrate these concepts, such as how funds that strategically adjust their investment styles based on changing market conditions may outperform those that strictly adhere to their original strategies. We have also included the latest research findings (Li and Jin, 2021; Koenig and Burghof, 2022; Yi et al., 2024; Liu and Yi, 2024; Zhang and Lv, 2024) to strengthen the theoretical background and support the research questions and objectives.

• The novelty including the main contributions of the study is missing. The introduction which serves as the bedrock of the study is not what I am expecting to see in a top journal like this.

Response: Thank you for your feedback. We have restructured the innovation points and main contributions of the paper. The specific content is as follows:

This paper may make contributions in the following aspects. Firstly, while prior literature focuses on the overall impact of fund style drift, this study differentiates the effects of distinct types of fund style drift behaviors on fund performance. Thi

---

## [Decision Letter · Decision Letter 1]

17 Jan 2025

PONE-D-24-36296R1Fund Style Drift and Fund Performance: Evidence from ChinaPLOS ONE

Dear Dr. Wei,

Thank you for submitting your manuscript to PLOS ONE. After careful consideration, we feel that it has merit but does not fully meet PLOS ONE’s publication criteria as it currently stands. Therefore, we invite you to submit a revised version of the manuscript that addresses the points raised during the review process.

I am writing regarding the recent changes made to your manuscript after it was accepted for publication. While we understand and appreciate your efforts to ensure the manuscript complies with the journal’s technical requirements, we noticed a significant reduction in the number of references from 57 to 32. This adjustment has raised concerns, as it substantially alters the context and scholarly foundation upon which the reviewers based their recommendations for acceptance. For example, in Section 4.4.2, the two references in the second paragraph were completely removed and this alters the context of the paper. Several other references were also removed in different sections of the paper without reason. The references cited in the reviewed version of the manuscript are integral to the argumentation, rigour, and scholarly value of your work. Their removal risks distorting the content that the reviewers and editorial team evaluated. To resolve this matter, we kindly request that you reinstate all references included in the reviewed version of the manuscript. If this adjustment is not feasible, the manuscript will need to undergo a new review process to ensure its scholarly integrity remains uncompromised.

We look forward to receiving your revised manuscript.

Kind regards,

Kingstone Nyakurukwa

Academic Editor

PLOS ONE

Journal Requirements:

Additional Editor Comments :

I am writing regarding the recent changes made to your manuscript after it was accepted for publication. While we understand and appreciate your efforts to ensure the manuscript complies with the journal’s technical requirements, we noticed a significant reduction in the number of references from 57 to 32. This adjustment has raised concerns, as it substantially alters the context and scholarly foundation upon which the reviewers based their recommendations for acceptance. For example, in Section 4.4.2, the two references in the second paragraph were completely removed and this alters the context of the paper. Several other references were also removed in different sections of the paper without reason. The references cited in the reviewed version of the manuscript are integral to the argumentation, rigour, and scholarly value of your work. Their removal risks distorting the content that the reviewers and editorial team evaluated. To resolve this matter, we kindly request that you reinstate all references included in the reviewed version of the manuscript. If this adjustment is not feasible, the manuscript will need to undergo a new review process to ensure its scholarly integrity remains uncompromised.

Reviewers' comments:

Reviewer's Responses to Questions

**Comments to the Author**

1. If the authors have adequately addressed your comments raised in a previous round of review and you feel that this manuscript is now acceptable for publication, you may indicate that here to bypass the “Comments to the Author” section, enter your conflict of interest statement in the “Confidential to Editor” section, and submit your "Accept" recommendation.

Reviewer #2: All comments have been addressed

2. Is the manuscript technically sound, and do the data support the conclusions?

Reviewer #2: Yes

3. Has the statistical analysis been performed appropriately and rigorously? 

Reviewer #2: Yes

4. Have the authors made all data underlying the findings in their manuscript fully available?

Reviewer #2: Yes

5. Is the manuscript presented in an intelligible fashion and written in standard English?

Reviewer #2: Yes

6. Review Comments to the Author

Reviewer #2: Comments have been addressed and that i recommend that the paper must be accepted and published in this journal

7. PLOS authors have the option to publish the peer review history of their article (what does this mean?). If published, this will include your full peer review and any attached files.

Reviewer #2: **Yes: **Michael Appiah

---

## [Author Response · Author response to Decision Letter 1]

18 Jan 2025

Thank you for your detailed feedback and for providing me with the opportunity to revise my manuscript titled “Fund Style Drift and Fund Performance: Evidence from China.” I appreciate the constructive comments from you and the reviewers, which have helped me improve the quality and rigor of my work.

I have carefully reinstated all references that were part of the reviewed version of the manuscript and addressed your concerns regarding the scholarly integrity of the paper. Additionally, I have ensured that the manuscript complies fully with the journal’s technical requirements.

As requested, I have uploaded the following files via the submission system:

 1. Revised Manuscript with Track Changes: Highlights all changes made in response to the editorial feedback.

 2. Manuscript: A clean, unmarked version of the revised manuscript.

Please find these files named as:

 • Revised Manuscript2_Fund Style Drift and Fund Performance

 • Manuscript2_Fund Style Drift and Fund Performance

If there are any additional changes or requirements, please do not hesitate to let me know. Thank you again for your guidance and for considering this revision for publication.

Best regards

---

## [Editor Report · Decision Letter 2]

21 Jan 2025

PONE-D-24-36296R2Fund Style Drift and Fund Performance: Evidence from ChinaPLOS ONE

Dear Dr. Wei,

Thank you for submitting your manuscript to PLOS ONE. After careful consideration, we feel that it has merit but does not fully meet PLOS ONE’s publication criteria as it currently stands. Therefore, we invite you to submit a revised version of the manuscript that addresses the points raised during the review process.

Thank you for the changes made to the previous version of the article. However, I have noticed that the tracked version of the article includes some changes in Chinese. Could you kindly translate these changes into English so that I can make an informed assessment of the revisions?

We look forward to receiving your revised manuscript.

Kind regards,

Kingstone Nyakurukwa

Academic Editor

PLOS ONE
---

## [Author Response · Author response to Decision Letter 2]

22 Jan 2025

Dear editor,

Thank you for your thorough review and feedback on our manuscript. We sincerely apologize for the oversight in the previous submission, which contained some unintended Chinese text.

We have carefully reviewed and revised the manuscript to ensure all content is in English. A new file, titled "Revised Manuscript3_Fund Style Drift and Fund Performance," has been uploaded, and the previous erroneous document has been deleted. The latest revised version no longer contains any Chinese text.

We appreciate your patience and understanding and look forward to your further comments.

Best regards.

---

## [Editor Report · Decision Letter 3]

28 Jan 2025

Fund Style Drift and Fund Performance: Evidence from China

PONE-D-24-36296R3

Dear Dr. Wei,

We’re pleased to inform you that your manuscript has been judged scientifically suitable for publication and will be formally accepted for publication once it meets all outstanding technical requirements.

Kind regards,

Kingstone Nyakurukwa

Academic Editor

PLOS ONE
---

## [Editor Report · Acceptance letter]

28 Jan 2025

PONE-D-24-36296R3 

PLOS ONE

Dear Dr. Wei, 

I'm pleased to inform you that your manuscript has been deemed suitable for publication in PLOS ONE. Congratulations! Your manuscript is now being handed over to our production team.

Kind regards, 

on behalf of

Mr. Kingstone Nyakurukwa 

Academic Editor

PLOS ONE